# A task-general connectivity model reveals variation in convergence of cortical inputs to functional regions of the cerebellum

**Maedbh King[1]\*[†], Ladan Shahshahani[2][†], Richard B Ivry[1,3], Jörn Diedrichsen[2,4,5]**

[1]Department of Psychology, University of California, Berkeley, Berkeley, United States; [2]Western Institute for Neuroscience, Western University, London, Canada; [3]Helen Wills Neuroscience Institute, University of California, Berkeley, Berkeley, United States; [4]Department of Statistical and Actuarial Sciences, Western University, London, Canada; [5]Department of Computer Science, Western University, London, Ontario, Canada

**Abstract** While resting-state fMRI studies have provided a broad picture of the connectivity between human neocortex and cerebellum, the degree of convergence of cortical inputs onto cerebellar circuits remains unknown. Does each cerebellar region receive input from a single cortical area or convergent inputs from multiple cortical areas? Here, we use task-based fMRI data to build a range of cortico-cerebellar connectivity models, each allowing for a different degree of convergence. We compared these models by their ability to predict cerebellar activity patterns for novel Task Sets. Models that allow some degree of convergence provided the best predictions, arguing for convergence of multiple cortical inputs onto single cerebellar voxels. Importantly, the degree of convergence varied across the cerebellum with the highest convergence observed in areas linked to language, working memory, and social cognition. These findings suggest important differences in the way that functional subdivisions of the cerebellum support motor and cognitive function.

## Editor's evaluation

This study presents the valuable finding that the human cerebellum receives highly convergent (rather than sparse) task-related connectivity from the cortex. A compelling case is made that this convergent connectivity supports a wide variety of cognitive functions, though it will be important for future work to fully verify the cortical-to-cerebellar directionality of the results. The work will be of broad interest to cognitive neuroscientists interested in the role of connectivity in cognition.

**\*For correspondence:**
maedbhking@gmail.com

[†]These authors contributed equally to this work

## Introduction

The last 30 years has witnessed a paradigm shift with regard to cerebellar function, with broad recognition that this subcortical structure is engaged in many aspects of human cognition. Since the first report of cerebellar activation in a semantic retrieval task (*Petersen et al., 1989*), thousands of neuroimaging papers have reported cerebellar recruitment during a broad range of tasks that cannot be attributed to the motor demands of these tasks. Functional interpretations include hypotheses concerning how the cerebellum may facilitate attentional shifts (*Allen et al., 1997*), stimulus-response mapping (*Bischoff-Grethe et al., 2002*), higher order rule processing (*Balsters et al., 2013*), verbal working memory (*Marvel and Desmond, 2010*), language (*Fiez, 2016*), and social cognition (*Van Overwalle et al., 2015*). This body of work has produced functional maps of the cerebellum that

depict the association of particular cognitive processes with different subregions of the cerebellum (*King et al., 2019*).

Given the relative uniform cytoarchitecture of the cerebellar cortex, it is assumed that differences in function mainly arise from variation in the input to the cerebellum. Trans-synaptic tracing methods employed in non-human primates studies have revealed extensive reciprocal connections between many frontal and parietal areas and the cerebellum (*Dum and Strick, 2003*; *Kelly and Strick, 2003*). These studies have highlighted the closed-loop nature of these connections, with each (neo-)cortical region projecting to a specific cerebellar region, and receiving input from the same area (*Strick et al., 2009*). In humans, resting state functional connectivity analyses have revealed a set of cerebellar networks, each one associated with a specific cortical network (*Buckner et al., 2011*; *Ji et al., 2019*; *Marek et al., 2018*).

An important unanswered question is whether each cerebellar region receives input from a restricted cortical region or whether it receives convergent input from multiple cortical regions. Providing an answer to this question has important implications for our understanding of cerebellar function. An architecture marked by a one-to-one relationship between cortical and cerebellar regions would suggest that the function of each cerebellar region is to fine-tune the dynamics in its cortical input. In contrast, a convergent architecture would suggest that subregions within the cerebellum integrate information across disparate cortical regions and may coordinate their interactions. Indeed, recent work in the rodent brain has suggested convergence of mossy fibers from diverse sources onto the same cerebellar region (*Henschke and Pakan, 2020*; *Pisano et al., 2021*), or even onto the same granule cells (*Huang et al., 2013*). Furthermore, the pattern of cortico-cerebellar convergence may vary across the cerebellar cortex, similar to how cortical areas show considerable variation in the degree to which they serve as points of convergence from other cortical regions (*Bertolero et al., 2015*; *Yeo et al., 2014*; *Yeo et al., 2015*).

In the current study, we introduce a novel approach to study cortico-cerebellar connectivity. Using the data from a large battery of tasks, we derived models that could be used to predict the activity in each cerebellar voxel based on the activity pattern in the neocortex. While this approach allowed us to evaluate the degree of convergence of cortical inputs to the cerebellar cortex, we recognize that the model could also be evaluated in the opposite direction, namely, to predict activity in the neocortex based on cerebellar activity patterns. However, we believe that using the model to make directional predictions from neocortex to cerebellum is most appropriate for fMRI data. The BOLD signal in the cerebellar cortex overwhelmingly reflects cortical input (via the pons) with no measurable contribution from the Purkinje cells, the output neurons of the cerebellar cortex (*Mathiesen et al., 2000*; *Thomsen et al., 2004*; *Alahmadi et al., 2016*; *Alahmadi et al., 2015*; *Mapelli et al., 2017*; *Gagliano et al., 2022*). In contrast, the neocortical BOLD signal reflects many sources including inputs from other cortical regions, local activity (input and output), as well as ascending input. While the latter will include cerebellar input via the thalamus, this source is likely to make a relatively small contribution to the overall BOLD response in the neocortex. As such, the relationship between neocortical and cerebellar BOLD signals will be most informative in evaluating cortico-cerebellar connectivity.

We trained multiple models of neocortical-cerebellar connectivity on a fMRI data set obtained while human participants completed a large task battery that was designed to engage cognitive processes across a broad range of functional domains (e.g. visual cognition, memory, attention, cognitive control). The models varied in terms of the degree of convergence of cortical inputs onto each cerebellar area. To evaluate the models in a cross-validated fashion, we examined how well each model predicted cerebellar data obtained from different tasks and/or different participants, using only the corresponding neocortical data. These analyses reveal a novel picture of cortico-cerebellar connectivity, one in which the degree of convergence was higher in cerebellar regions associated with more complex cognitive functions.

## Results

### Overview

We compared three models of cortico-cerebellar connectivity by using a region-to-region predictive modeling approach (*Cole et al., 2016*; *Mell et al., 2021*). For each model, the task-evoked activity in each cerebellar voxel was predicted as a linear combination of task-evoked activity patterns across

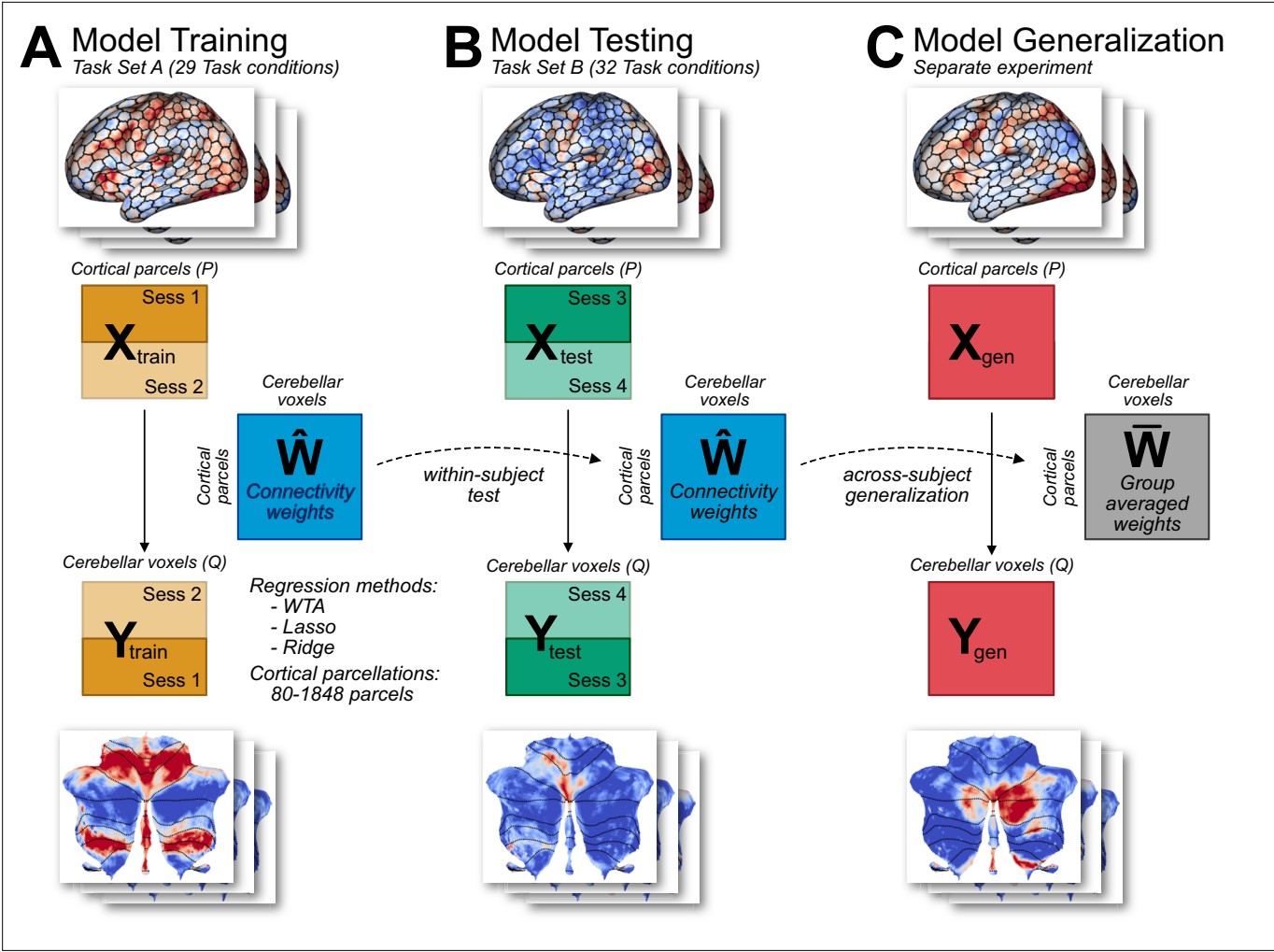

**Figure 1.** Connectivity model training and evaluation. (**A**) Models were trained on Task Set A (session 1 and 2) of the multi-domain task battery (MDTB; *King et al., 2019*). Model hyperparameters were tuned using fourfold cross validation. Three types of models were used (WTA, Lasso, Ridge), each with 7 cortical parcellations of different granularity. (**B**) Models were tested on an independent Task Set B (sessions 3 and 4), which included both novel and common tasks. Models had to predict the cerebellar activity solely from cortical activity patterns. To avoid the influence of shared noise correlations across cortex and cerebellum, the models were trained and tested by using cortical and cerebellar activity patterns from different sessions within each task set (see Methods). (**C**) As a test of generalization, the models were used to predict cerebellar activity from cortical data obtained from a separate experiment (King et al., unpublished data).

the entire neocortex (*Figure 1*). As a model of sparse connectivity, we used a Winner-Take-All (WTA) model which imposes the strong constraint that only a single cortical region is used to predict the activity in each cerebellar voxel. The other two models allowed for some degree of convergence. Models estimated with Lasso Regression (L1 regularization) find sparse solutions, minimizing the number of cortical inputs by setting to zero those weights that make a negligible contribution to the predicted activity. Models estimated with Ridge Regression (L2 regularization) allow for a wide distribution of inputs, keeping each weight as small as possible. For model input, we used a set of cortical parcellations that varied in terms of their level of granularity. We trained the models on cortical and cerebellar fMRI data obtained from 24 participants who performed a task battery with 29 task conditions (Task Set A, *Figure 1A*). These were acquired over two sessions on separate days, with the tasks identical across sessions. These data were used to estimate the connectivity weights separately for each model and participant (see Methods).

To compare the models, we tested their prediction performance on two independent datasets. First, we used data from the same 24 participants when tested in two different sessions (Task Set B) that included 18 novel conditions and 14 conditions repeated from Task Set A (*Figure 1B*). To predict

the cerebellar activity patterns, we used the observed cortical activity patterns from Task Set B and the estimated connectivity weights () from Task Set A for each participant. Note that because the model predictions relied on a single set of connectivity weights across all tasks, the input is based only on the cortical activity patterns without reference to any features of the tasks themselves. Second, we also tested how the models would generalize when tested with data from a new group of participants tested on a set of novel tasks (*Figure 1C*).

The use of separate training and evaluation datasets allowed us to determine the best task-general model of cortico-cerebellar connectivity without overfitting the data. To validate that this approach enabled us to distinguish between different forms of cortico-cerebellar connectivity, we conducted a range of model recovery simulations (see Methods for details). Using the measured cortical activity for each participant, we generated artificial sets of cerebellar data imposing either a one-to-one or a many-to-one mapping. Following the procedure used with the real data, we trained the three models on Task Set A and tested them on Task Set B. The simulations (*Figure 2—figure supplement 1*) showed that the WTA performed best if each cerebellar voxel was connected to only one cortical parcel, whereas ridge regression performed better if there was substantial convergence, with Lasso performing at an intermediate level. Thus, despite the presence of some degree of collinearity between the cortical parcels, these simulations demonstrate that our modeling approach is able to distinguish between different forms of connectivity.

## Cortico-cerebellar connectivity is best captured by models with convergence

We first compared the different models, asking how well they predicted activity patterns obtained when the same participants were tested on Task Set B (*Figure 2A*). Models allowing for some degree of convergence outperformed the WTA model, and this advantage was observed across all levels of cortical granularity. Indeed, the prediction performance for the Ridge, Lasso, and WTA models was relatively independent of granularity. Averaged across all levels of granularity, the Ridge models outperformed the WTA models ($F_{1,23}$ = 47.122, p<0.001). Post-hoc tests revealed that this advantage for the Ridge model was consistent across all levels of granularity, starting with a parcellation of 80 regions ($t_{23}$=5.172, p<0.001). We also found a significant difference in predictive accuracy between the Ridge and Lasso model ($F_{1,23}$ = 15.055, p<0.001). Post-hoc tests showed that there was no significant difference in predictive accuracy at the lowest level of granularity ($t_{23}$=1.279, p=0.213), whereas the difference was significant for the finer parcellations (all $t_{23}$ >10.937, p<0.001).

The mean predictive accuracy of the Lasso and Ridge models was 0.257 (*Figure 2A*). There are two issues of note here. First, the models predicted activity of individual voxels, without any smoothing across cerebellar voxels. Thus, the upper bound for the predictive accuracy is limited by the reliability of the cerebellar test data. The correlation of the measured cerebellar activity patterns across the two sessions of Task Set B was, on average, r=0.51 (SD = 0.102). Reliability was fairly consistent across the cerebellum (*Figure 2C*), with some decreases in lobules I-IV. This dropoff likely reflects the fact that our battery only included hand and eye movements, and did not activate the lower body representation that is prominent in this area. Second, predictive accuracy is also limited by the reliability of the cortical data that was used to make the prediction. This is an especially limiting factor for WTA models that use fine granularity, since the predictions will be based on data obtained from a small cortical region.

Given these issues, we calculated a noise ceiling for each model, taking into account the reliability of the cerebellar data, the reliability of the cortical data, and the effect of granularity (see Methods, *Figure 2—figure supplement 2*). As an unbiased comparison of the model predictions, *Figure 2B* re-plots the predictive accuracy of each model normalized by its noise ceiling. As with the original analysis, the Ridge model significantly outperformed the WTA model ($F_{1,23}$ = 16.49, p<0.01), and in post-hoc tests, the advantage was especially pronounced for finer parcellations (1848 regions; $t_{23}$=5.073, p<0.001). Overall, the noise ceiling calculation showed that the Ridge model was able to predict approximately 45% of the systematic variance of the cerebellar activity patterns across tasks (average $R^2$=$0.67^2$). While the predictive accuracy (*Figure 2D*) was best in anterior motor regions, it was reasonably high across the entire cerebellar surface.

The predicted and observed activity patterns for two exemplary tasks (*Figure 2E*) demonstrate the quality of these predictions at the group level. In both of these examples, (spatial working memory

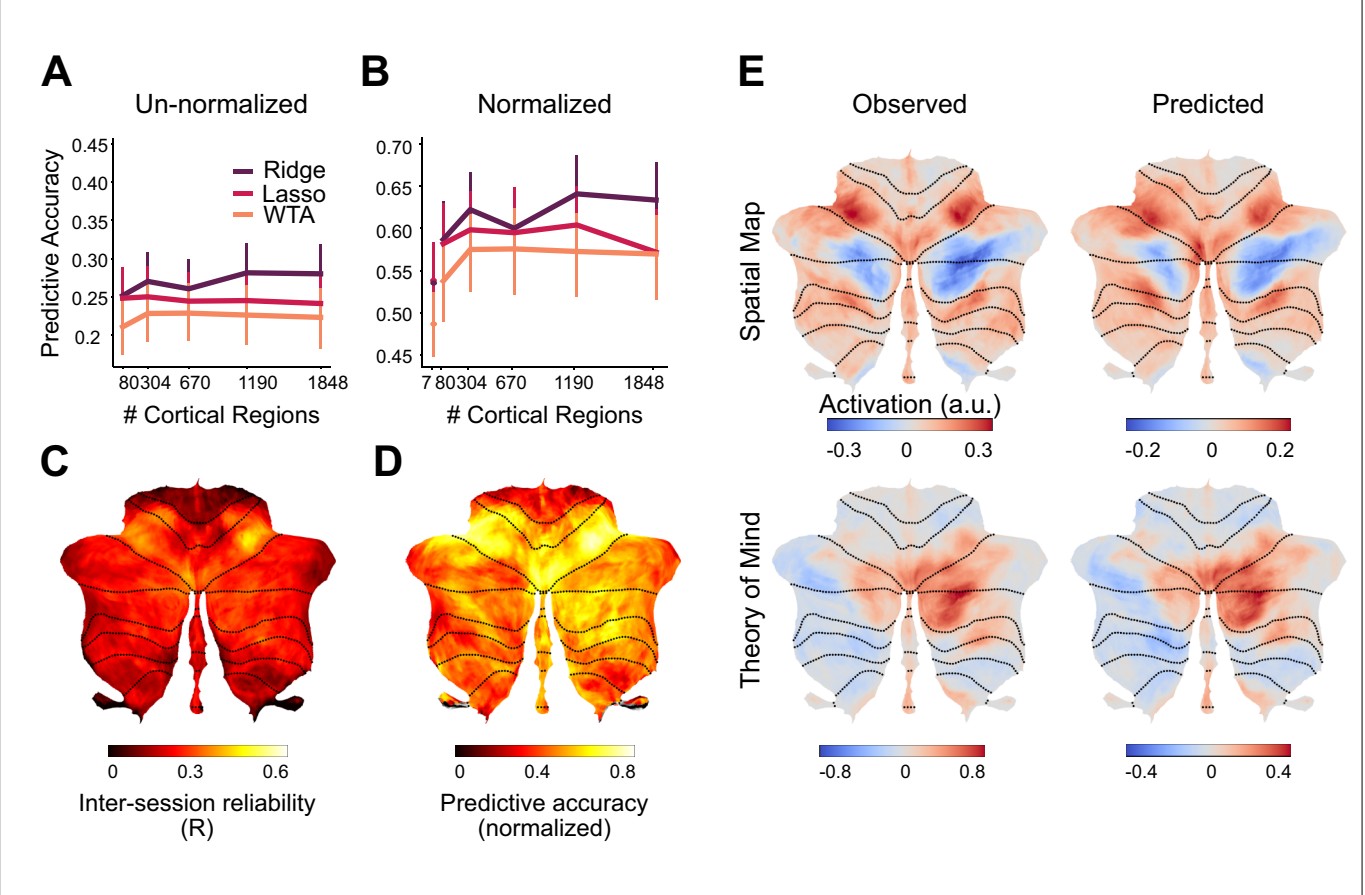

**Figure 2.** Performance of cortico-cerebellar connectivity models. (**A**) Predictive accuracy (Pearson correlation) of the Ridge, Lasso, and WTA regression models for the test data of Task Set B (**B**) Predictive accuracy normalized to the noise ceiling based on reliability of both cerebellar and cortical data (see Methods). Source data files are available for panels A and B ('model_evaluation.csv') (**C**) Voxelwise map of inter-session reliability of the test data. (**D**) Voxelwise map of predictive accuracy of the Ridge model (1848 parcels), normalized to the noise ceiling. (**E**) Observed and predicted activity for a novel task involving spatial working memory (Spatial Map) and a social cognition task (Theory of Mind). The spatial map task was not included in Task Set A. Error bars indicate 95% confidence intervals across participants (n=24).

The online version of this article includes the following source data and figure supplement(s) for figure 2:

**Source data 1.** Source data file contains model evaluation predictive accuracies for each of the three methods (Ridge, Lasso, Winner-Take-All), and for each parcellation (80-1848).

**Figure supplement 1.** Model recovery simulations demonstrate the ability to identify different forms of cortico-cerebellar connectivity.

**Figure supplement 2.** Noise ceiling for Ridge model.

**Figure supplement 3.** Predictive accuracy for Ridge and WTA models using functional cortical parcellations.

**Figure supplement 4.** Hyper-parameter tuning for connectivity models.

**Figure supplement 4—source data 1.** Source data file contains predictive accuracies for each hyperparameter (-2, 0, 2, 4, 6, 8, 10) of the Lasso model, and for each parcellation (80-1848).

**Figure supplement 4—source data 2.** Source data file contains predictive accuracies for each hyperparameter (-5, -4, -3, -2, -1) of the Ridge model, and for each parcellation (80-1848).

and social cognition tasks), the connectivity model predicted the pattern of task activity with a high degree of fidelity. This is especially compelling for the spatial working memory task as the training set did not include a task with similar characteristics.

To ensure that our results were not biased by the use of an essentially arbitrary parcellation of the neocortex, we repeated the analysis using a range of published functional parcellations. For example, we used a 7-network cortical parcellation based on resting state fMRI data (*Yeo et al., 2011*) to train and test the three models. Here, too, the WTA model was inferior to the Ridge models ($t_{23}$=2.956,

p<0.01), with no performance difference between Ridge and Lasso models ($t_{23}$=−1.235, p=0.229). The same pattern held when we used other common functional parcellations of the neocortex (*Figure 2—figure supplement 3*).

In summary, these results demonstrate that models which entail some degree of convergence from the neocortex to the cerebellum outperform a model in which cerebellar activity is based on input from a single cortical region. This conclusion holds across a broad range of cortical parcellations (*Zhi et al., 2022*).

## Convergence of neocortical inputs varies across the cerebellum

To gain insight into where these cortical inputs came from, we visualized the cortical weights for each of the 10 functional regions of the cerebellum (*Figure 3—animation 1*). As expected given the crossed connectivity between M1 and lobules IV/V (*Kelly and Strick, 2003*; *Krienen and Buckner, 2009*; *O'Reilly et al., 2010*), the input to the hand regions of the cerebellum (regions 1 and 2) was centered

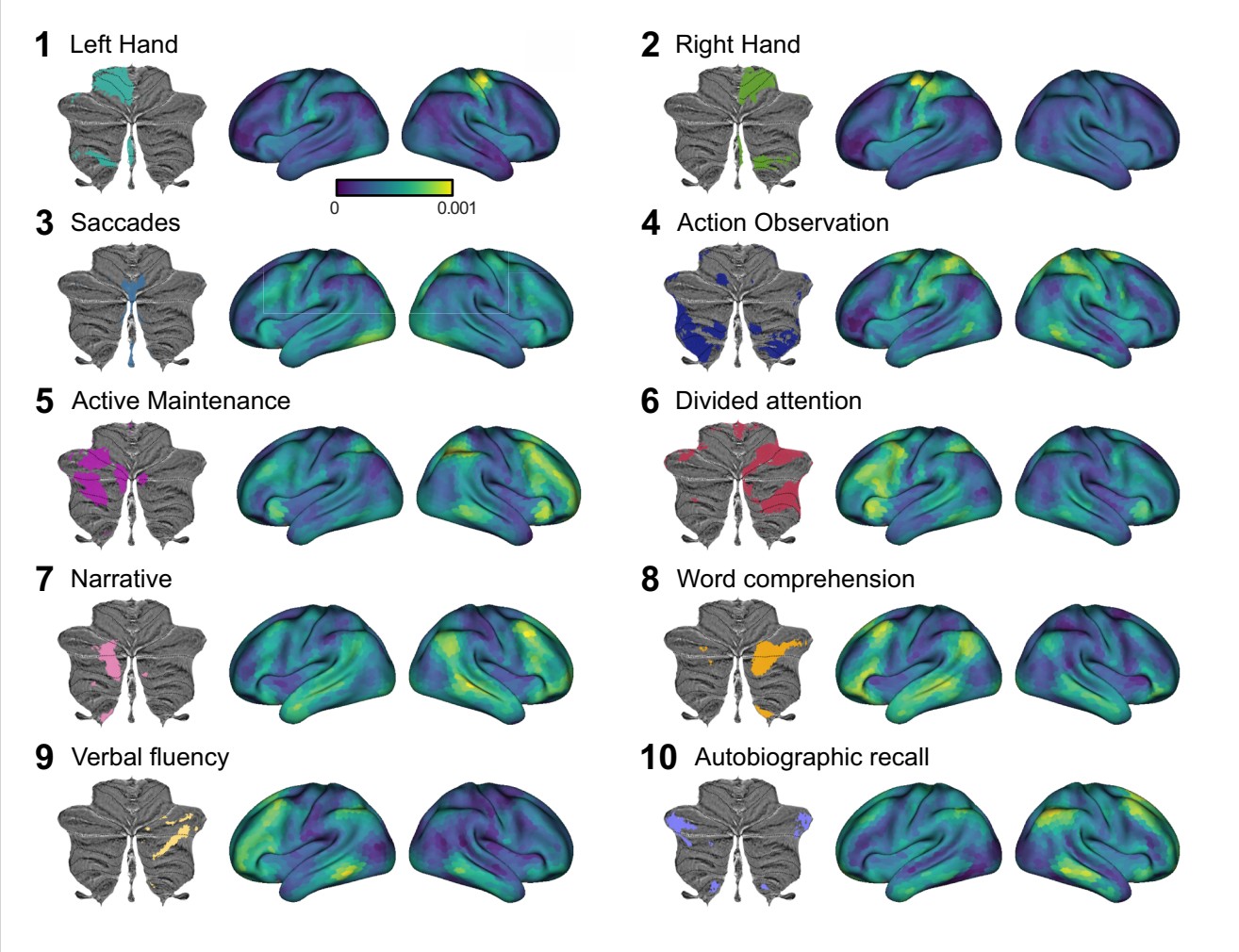

**Figure 3.** Cortical connectivity weight maps for the Ridge regression model with 1848 cortical parcels for each of 10 functional cerebellar regions. Each region is denoted by the most important functional term (*King et al., 2019*). Results are averaged across participants. Regression weights are in arbitrary units. See *Figure 3—animation 1* for a gif of the connectivity weight maps, and see *Figure 3—figure supplement 1* for the corresponding analysis using Lasso regression. Figure 3 has been adapted from Figure 5 from *King et al., 2019*.

The online version of this article includes the following video and figure supplement(s) for figure 3:

**Figure 3—animation 1.** Animated cortico-cerebellar connectivity maps for the Ridge regression model.

**Figure 3—animation 2.** Animated cortico-cerebellar connectivity maps for the Lasso regression model.

**Figure supplement 1.** Cortical connectivity weight maps for the Lasso model with 1848 cortical regions.

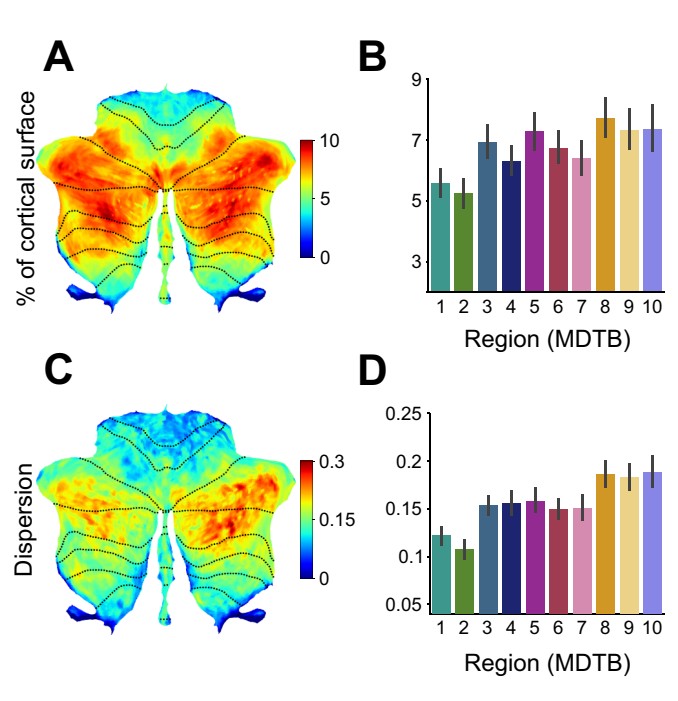

**Figure 4.** Cortico-cerebellar convergence measures using the Lasso model. (**A**) Map of the cerebellum showing percentage of cortical parcels with non-zero weights for the Lasso model (n=80 parcels). (**B**) Percentage of parcels with non-zero weights for functional subregions of the cerebellum. (**C**) Spherical dispersion of the connectivity weights on the cortical surface for each cerebellar voxel. (**D**) Average cortical dispersion for each functional subregion of the cerebellum. Error bars indicate standard error of the mean across participants. See *Figure 4—figure supplement 1* for the same results using Ridge regression. Error bars indicate 95% confidence intervals across participants (n=24).

The online version of this article includes the following figure supplement(s) for figure 4:

**Figure supplement 1.** Cortico-cerebellar convergence measures using the Ridge model.

**Figure supplement 2.** Percentage of cortical surface across levels of granularity for Lasso regression model.

around the contralateral primary sensorimotor cortex with some additional input from premotor and parietal cortex. Regions 3 and 4 are the other two cerebellar regions associated with motor function. Activity in region 3, the oculomotor vermis, is predicted by a bilateral set of cortical regions including the frontal eye fields (FEF), regions in the intraparietal sulcus, and extrastriate visual regions. Region 4, an area strongly activated during action observation, is predicted by a bilateral network of regions including premotor cortex, supplementary motor cortex (SMA) and parietal cortex. Cerebellar regions 5–10, the regions associated with more cognitive processes are predicted by a distributed set of cortical regions, with stronger input coming from the contralateral cortical hemisphere. For example, cerebellar region 5, a region restricted to the left cerebellum, receives much stronger input from the right cerebral hemisphere. In summary, these results suggest that most cerebellar regions are best predicted by multiple cortical regions, pointing to some degree of cortical-cerebellar convergence.

Visual inspection of the connectivity patterns (*Figure 3*) also highlights interregional variation of convergence. For example, inputs to hand motor regions MDTB (regions 1 and 2) arise from a relatively small area of the neocortex, while inputs to regions in lobule VII (regions 3–10) come from a larger area. To quantify this observation, we tallied the number of non-zero regression weights for each cerebellar voxel as a measure of its input surface area. For this calculation, we used the Lasso model (80 cortical parcels) since it uses the minimal set of cortical areas necessary for predicting each cerebellar voxel. As the Lasso model forces the estimates of the other weights to zero, it allows for a quantification of input area without applying an arbitrary threshold. This calculation revealed a substantial variation in the degree of convergence across the cerebellar cortex (*Figure 4A*). For example, predicting the activity pattern of voxels within Crus I required inclusion of up to 10% of the

cortical surface whereas predicting the activity of voxels in the anterior lobe required less than 5% of the neocortex.

To statistically analyze these data, we opted to bin the cerebellar data using a functional 10-region parcellation of the cerebellum (*King et al., 2019*). This analysis confirmed that the size of the estimated cortical input area differed across functional regions ($F_{9,207}$ = 7.244, p<0.001, *Figure 4B*). The lowest level of convergence was observed for regions 1 and 2, the anterior hand regions of the cerebellum. The highest levels of convergence were found in regions of the cerebellum associated with more cognitive functions (e.g. regions 7 and 8, areas engaged during language comprehension). Noteworthy, this pattern holds for cortical parcellations of different levels of granularity (*Figure 4—figure supplement 2*), as well as for the thresholded coefficients from the Ridge regression models (*Figure 4—figure supplement 1A, B*).

To assess whether these differences were driven by the collinearity of the cortical data, rather than by cortical-cerebellar connectivity, we ran a simulation (see Methods, model recovery simulations) in which we replaced the cerebellar data with the activity profile of the most similar cortical parcel. In this simulation, we did not find any differences in the area of estimated input across cerebellar regions ($F_{9,207}$=1.762, p=0.076). Thus, the observed variation in convergence cannot be explained by the collinearity between different cortical input regions.

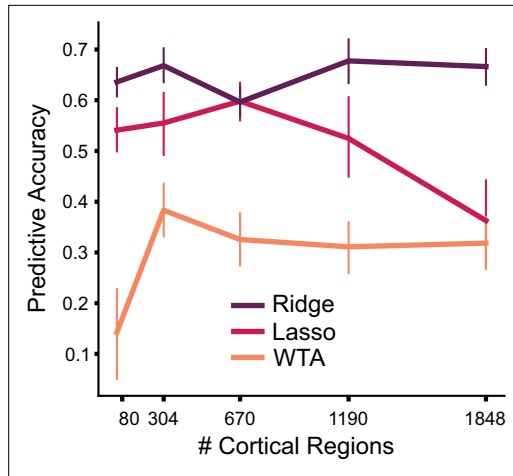

**Figure 5.** Generalization to new dataset. Models of cortico-cerebellar connectivity are tested in a new experiment. Each model is tested across different levels of cortical granularity. Predictive accuracy is the Pearson correlation between observed and predicted activity patterns, normalized to the noise ceiling. Error bars indicate 95% confidence intervals across participants (n=20).

As an independent method to quantify convergence, we determined the spatial spread of the inputs across the cortical surface. For example, MDTB region 4, which is activated by action observation, is explained by a set of cortical regions with a relatively small surface area. Nonetheless these regions are spread widely across the cortex (e.g. fusiform gyrus, parietal and premotor regions). For a measure of dispersion, we calculated the spherical variance of the non-zero connectivity weights on an inflated model for each cerebral hemisphere (see Methods). This analysis revealed a similar pattern as seen in the surface area measure (*Figure 4C, D* $F_{9,207}$=18.322, p<0.001). For example, the cortical inputs to the hand motor regions of the cerebellum were more concentrated, whereas the cortical inputs to lobule VII were more dispersed. Again, this pattern also holds for the Ridge models (*Figure 4—figure supplement 1C, D*).

In summary, we observed variation in the degree of cortico-cerebellar convergence across the cerebellar cortex. In particular, cerebellar motor areas such as the hand region in lobule V and VIII received relatively focal cortical input whereas cerebellar areas associated with cognitive processes (e.g. language, working memory) in lobule VII exhibited a higher degree of convergence.

## Cortico-cerebellar connectivity model predicts new cerebellar data

A strong test of a model is how well it predicts novel data obtained in contexts distinct from that data used to develop the model. We conducted a generalization test using data from a separate experiment involving novel tasks and new participants. The data came from a study in which participants were trained over multiple sessions on five tasks, selected to span a set of distinct cognitive domains.

All three model types (WTA, Lasso, Ridge) were evaluated, using the connectivity weights estimated from Task Set A of the main study. Because the new study had different participants, we averaged the weights for each model across the individuals in the training set. These group-averaged weights were then used to predict cerebellar activity patterns for the new data.

As shown in *Figure 5*, the connectivity models were generally successful in predicting cerebellar activity patterns for the new dataset. The overall pattern is quite similar to that seen in the initial

model tests (in which we had used data from different tasks) but involved training and test data from the same participants. Predictive accuracy was stable across levels of cortical granularity and the Ridge model provided the best overall predictive accuracy ($r=0.657$) and the WTA ($r=0.352$) the worst predictive accuracy. These results provide evidence that our cortico-cerebellar connectivity models capture important aspects of connectivity that are stable both across tasks and participants. Moreover, in accord with the earlier analyses based on predictions at the individual level, this generalization analysis again suggests that approximately 43% of the variation of cerebellar activity across tasks can be predicted by cortical activity alone.

## Discussion

To date, models of connectivity between the human neocortex and cerebellum have been based on fMRI resting-state data (*Buckner et al., 2011*; *Ji et al., 2019*; *Marek et al., 2018*). This work demonstrates that each region of the cerebellum receives input from a distinct set of cortical regions. For example, anterior and posterior cerebellar motor regions show correlated activity with contral-lateral sensorimotor cortex and non-motor or 'cognitive' cerebellar regions show correlated activity with specific parietal and frontal association areas.

Despite these important insights, previous work has not been designed to examine the patterns of convergence between the neocortex and cerebellum. Resting state connectivity maps are generally produced by assigning each cerebellar voxel to a single cortical network, a *de facto* winner-take-all model. In the present study, we quantified and compared models of cortico-cerebellar connectivity. The models differed in the degree of convergence of cortical inputs to each cerebellar region, ranging from an architecture constrained to a strict one-to-one mapping to architectures that allowed for distributed inputs. We evaluated these models in terms of how well they could predict cerebellar activity patterns on a novel Task Set. For nearly the entire cerebellum, models that allowed for some convergence predicted cerebellar activity better than the WTA model.

### Convergence differs across cerebellar circuits

Importantly, the amount of convergence differed across the cerebellar cortex. Specifically, regions in anterior (lobules I-V) and inferior cerebellum (lobules VIII-X) were well predicted by a relatively small and concentrated area of the neocortex. In contrast, regions in lobule VI and especially lobule VII required input from a larger and spatially more dispersed set of cortical regions that were primarily located in association areas of prefrontal, parietal, and cingulate cortex. This finding underscores the heterogeneity of cortico-cerebellar connectivity with some cerebellar areas functioning in nearly a 1:1 relationship with a single cortical region, whereas other areas integrate input from a more diverse set of cortical regions.

This variation bears some resemblance to a motor/cognitive gradient identified from resting state data (*Guell et al., 2018*). However, there are a few notable exceptions. First, based on our evaluation metrics, Region 3 (oculomotor vermis) was best explained by a large and relatively dispersed set of cortical regions, including intraparietal sulcus, the frontal eye fields (FEF), and extrastriate visual areas (*Figure 4*). Second, sub-regions of lobules IX, functionally associated with non-motor tasks (*Buckner et al., 2011*), are best explained by a relatively restricted set of cortical regions (*Figure 4A and C*). Thus, rather than following a functional distinction, it appears that regions with strong convergence are anatomically restricted to lobules VI and VII, two areas that have disproportionately increased in size during primate evolution (*Balsters et al., 2010*).

Variation in cortico-cerebellar connectivity has important implications for theories of cerebellar function (*De Zeeuw et al., 2021*; *Diedrichsen et al., 2019*; *Henschke and Pakan, 2020*). A cerebellar region that forms a closed-loop circuit with a single cortical region can only fine-tune the computational processes that are ongoing in its connected cortical region. In contrast, a cerebellar region that receives convergent input would be able to coordinate the interactions between dispersed cortical regions. Indeed, using a rodent model, *Pisano et al., 2021* have identified cerebellar areas that project back to multiple cortical areas, providing a substrate by which the cerebellum can influence neuronal dynamics not only in a focal cortical area but also interactions within networks of cortical areas (*Pisano et al., 2021*).

## Task-based vs. resting-state connectivity analyses

While our modeling approach could be applied to resting-state data, we opted to use the data from the multi-domain task battery for a variety of reasons. First, the breadth of tasks included in the battery assured that there would be reliable variation in activity across most of the neocortex and cerebellum, a necessary precondition for building a complete connectivity model. Second, the dataset allowed us to avoid biases that arise from correlated noise across abutting cortical and cerebellar regions (*Buckner et al., 2011*). We used the cortical activity patterns from one session to predict cerebellar activity patterns in a different session, relying on the fact that measurement noise is independent across sessions. Third, the MDTB data set allowed us to test the model across a broad set of tasks and mental states. While resting-state correlations are predictive of task-based activation patterns (*King et al., 2019*; *Tavor et al., 2016*; *Zhi et al., 2022*), resting in an fMRI scanner is arguably a relatively restricted situation. Our generalization test shows that the connectivity model can robustly predict activity patterns for new tasks and participants.

## Directionality of the model

We opted in the present study to model the correlations between neocortical and cerebellar activity in a directional manner, making inferences from these data about connectivity from the neocortex to the cerebellum. We recognize that the two structures communicate with each other in a closed loop. Our decision to focus on cortico-cerebellar connectivity is based on studies of cerebellar blood flow in rodents. Increases in cerebellar blood flow, a major contributor to the BOLD signal, are the result of increases in mossy fiber activity, the primary input to the cerebellar cortex (*Alahmadi et al., 2015*; *Alahmadi et al., 2016*; *Gagliano et al., 2022*; *Mapelli et al., 2017*; *Mathiesen et al., 2000*; *Thomsen et al., 2004*). In contrast, even a dramatic increase in the (complex or simple spike) firing rate of Purkinje cells does not produce a measurable change in blood flow within the cerebellar cortex (*Thomsen et al., 2004*; *Thomsen et al., 2009*). Thus, the cerebellar BOLD signal provides a relatively clear image of the cortical inputs to the cerebellum; importantly, this signal does not provide information about the output of the cerebellar cortex (or cerebellum in general). In contrast, the BOLD signal in the neocortex reflects a combination of signals including local inputs/outputs across cortical layers, inputs from other cortical regions, and ascending inputs such as from the thalamus which will include input from the cerebellum. Given this asymmetry, a priori, we should expect that correlations in the fMRI signal between the neocortex and cerebellum will more strongly reflect the flow of information from the cortex to the cerebellum than the reverse.

Nonetheless, it is possible that some of the convergent inputs identified in our modeling work may be due to divergent projections from a single cerebellar region to multiple cortical regions. Indeed, recent viral tracing work has shown that some cerebellar areas project to the reticular nuclei in the thalamus, which in turn projects to a wide array of cortical regions (*Kelly and Strick, 2003*; *Pisano et al., 2021*). A complete analysis of the entire cortico-cerebellar circuit will require the methods such as viral tracing techniques (*Kelly and Strick, 2003*; *Pisano et al., 2021*) or functional activation measures that target key nodes in the ascending pathway (deep cerebellar nuclei, thalamus).

## Methodological limitations

Inter-region correlations of fMRI data can of course only provide indirect evidence of true functional connectivity. As such, it is important to consider methodological limitations that may influence the validity of our conclusions. From a statistical perspective, it was not clear, a priori, that we would have the power to distinguish between models of connectivity given that there can be substantial collinearity between different cortical regions. The model-recovery simulations (*Figure 2—figure supplement 1A, B*) suggest that the present dataset was suitable to make such inferences, namely, activity patterns in different cortical regions were sufficiently de-correlated, likely reflecting the use of a broad task battery. Thus, we were able to recover the correct model used to simulate the data, regardless of whether we assumed one-to-one connectivity or different degrees of convergence.

However, the simulations also indicated that the approach (and data) was not sufficient to determine the absolute degree of convergence with high confidence. For example, the size of the cortical input area for each cerebellar region differed substantially between the Ridge and Lasso regression models (*Figure 3* vs. *Figure 3—figure supplement 1*). Nonetheless, the two models result in similar predictive accuracy when using real data. Therefore, the true extent of the cortical input to a cerebellar

region likely lies somewhere between the extremes provided by the Ridge and Lasso models. Importantly, this ambiguity does not impact the core observation that the degree of convergence varies systematically across the cerebellar cortex. Whether using measures based on cortical surface area or dispersion, the general picture of variation in connectivity holds for both the Ridge and Lasso models, as well as when using cortical parcellations of different granularity. Thus, we are confident that the variation in convergence reflects a stable, method-independent characteristic of the cortico-cerebellar system.

## Future directions

Our approach was designed to identify the best task-general model of cortico-cerebellar connectivity. The connectivity model was able to take cortical activity patterns to make fairly accurate predictions of cerebellar activity, even when the data were obtained in a completely separate experiment from that used to build the model. This finding suggests that a substantial proportion of the cortico-cerebellar communication can be described by fixed information flow (*Cole et al., 2016*); that is, each area of the cerebellum simply receives, independent of task, a copy of the neural activity occurring in the corresponding cortical areas.

However, the task-general model did not provide a perfect prediction of cerebellar activity. While these predictions may improve with more training data, we believe that there will likely be some systematic failures of the model. Such task-dependent deviations from the model prediction may offer important insights into cerebellar function, indicating that the cerebellum is more active than predicted by the cortical activity for some tasks and less for others. This pattern would suggest a 'gating' of cortical input to the cerebellum, perhaps within the pontine nuclei, which is the main relay station of cortical inputs to the cerebellum. Rather than just transmitting cortical input to the cerebellar cortex, these nuclei may serve as an adaptive gate (*Schwarz and Thier, 1999*), amplifying or attenuating information as a function of the relative importance of cerebellar processing for the current task.

The models presented here allow us to detect such deviations by comparing the observed cerebellar activity for any task against the activity that is predicted by a task-invariant connectivity model. As such, the model provides a potent new tool to test hypotheses concerning cerebellar function.

## Methods

### Multi-domain task battery

To build models of cortico-cerebellar connectivity, we used the publicly available multi-domain task battery dataset (MDTB; *King et al., 2019*). The MDTB includes fMRI data from two independent Task Sets (A and B). Each set consists of 17 tasks, eight of which were common to both sets. The 26 unique tasks were designed to sample activity during a broad range of task domains including motor (e.g. sequence production), working memory (e.g. 2-back task), language (e.g. reading), social (e.g. theory of mind), cognitive control (no-go, Stroop), and emotion (e.g. facial expression; see Table S1 in *King et al., 2019*).

Twenty-four participants (16 females, 8 males, mean age = 23.8) were scanned during four sessions, with Task Set A employed in sessions 1 and 2 and Task Set B in sessions 3 and 4. Within a session, there were eight 10-min runs, with each run composed of 17 blocks of 35 s each. Each block involved a unique task and included an initial 5 s instruction period. Most tasks contained multiple conditions (i.e. different levels of difficulty), resulting in a total of 47 unique task conditions.

### Image acquisition and preprocessing

All fMRI and MRI data were collected on a 3T Siemens Prisma located at the Center for Functional and Metabolic Mapping at Western University, Canada. The protocol used the following parameters: 1 s repetition time; field-of-view measuring 20.8 cm; P-to-A phase encoding direction; 48 slices; 3 mm thickness; in-plane resolution 2.5×2.5 mm$^2$. In order to localize and normalize the functional data, a high-resolution anatomical scan (T1-weighted MPRAGE, 1 mm isotropic resolution) of the whole brain was acquired during the first session.

Functional data were realigned for head motion artifacts within each session and different head positions across sessions using a six-parameter rigid body transformation. The mean functional image

was co-registered to the anatomical image and this rigid-body transformation was applied to all functional images. No smoothing or group normalization was applied. SPM12 ( spm/doc/spm12_manual. pdf) and custom-written scripts in MATLAB were used to conduct data pre-processing.

## General linear model (GLM)

To generate estimates of the activity related to each task condition, a general linear model (GLM) was fitted to the time series data of each voxel ($t_i$). This was done separately for each imaging run and Task Set (A and B). The Design matrix for the GLM (**Z1**) of the GLM consisted of the regressors for the different conditions, plus separate regressors for each of the 5-s task-specific instructions period. For each Task Set, the beta weight for each task condition was averaged across the 8 runs within a session, resulting in the average activity estimate for that session ($\bar{b}_j$) for each voxel. The regressor for Task Set A resulted in 46 activity estimates (17 Instructions + 29 condition regressors) per session and Task Set B resulted in 49 activity estimates (17 instructions + 32 conditions regressors) per session.

## Neocortex surface reconstruction

For each of the 24 participants, the anatomical surfaces of the cortical hemispheres were reconstructed using the standard recon-all pipeline of the FreeSurfer package (*Fischl, 2012*; v. 5.0). The pipeline included brain extraction, generation of white and pial surfaces, inflation, and spherical alignment to the symmetric fsLR-32k template (*Van Essen et al., 2012*). Individual surfaces were re-sampled to this standard grid, resulting in surfaces with 32,492 vertices per hemisphere.

## Spatial normalization of cerebellar data

The cerebellum was isolated and normalized to the high-resolution Spatially Unbiased Infratentorial Template of the cerebellum using the SUIT toolbox (*Diedrichsen, 2006*). This non-linear transformation was applied to both the anatomical and functional data. Task condition activity estimates (i.e. the beta weights) were resampled to a resolution of 3 mm isotropic and resliced into SUIT space. The cerebellar isolation mask was edited to remove voxels in the superior cerebellum that abutted voxels in the primary visual cortex. Functional images were masked with the cerebellar isolation mask resulting in activation signals that originate only from the cerebellar cortex. The cerebellar data were visualized using a surface-based representation of the cerebellar gray matter included in the SUIT toolbox (*Diedrichsen and Zotow, 2015*). This flat map is not intended to represent a true unfolding of the cerebellar cortex, but rather provides a convenient way to visualize volume-averaged cerebellar imaging data in a 2d representation.

## Connectivity models

All of the task-based connectivity models were, in essence, multiple-regression models in which the data for each of cerebellar voxels (yi) was modeled as a linear combination of Q cortical parcels (X, see next section).

$$y_i = Xw_i + e_i \tag{1}$$

By combining the data (Y), the connectivity weights (W), and the measurement error (E) for each cerebellar voxels as columns into a single matrix, the entire model could be written as:

$$Y = XW + E \tag{2}$$

The connectivity approach that we used relied only on task-evoked activity: model training and testing were done on the fitted time series rather than the full or residual time series (as done in many other connectivity approaches). We multiplied $Z_j$ from the first-level GLM with the 46 (for training) or 49 (for testing) activity estimates ($\bar{b}_j$) from the first-level GLM with the first-level design matrix (Z) to obtain the fitted time series the first-level design matrix for each session ($\bar{Z}_j$) from the first-level GLM with $t_{\text{fitted}i} = Z\bar{b} + E$ for each voxel (see *General Linear Model*). This was done for both the cortical and the cerebellar voxels.

In calculating the variances and covariances between cortical and cerebellar data (which are computed in the multiple regression models), this procedure reweighted the activity estimates to account for the fact that estimates for the instructions were based on 5 s of fMRI data, while estimates

for the conditions were based on 10–30 s of data. For example, the covariance between the cortical and cerebellar time series is:

$$Y^T X = \bar{b} X^T Z^T Z \bar{b} Y \qquad (3)$$

Therefore, if we had used the task-evoked activity estimates ($\bar{b}$) for cerebellar and cortical data, but reweighted the influence of each regressor by the diagonal of, $Z^T Z$ we would have obtained similar results. Because of the off-diagonal terms, this method also mitigates problems that may arise in event-related designs due to correlations between regressors (*Mumford et al., 2012*), one example is the estimation covariance between the instruction period and the task-related regressor that follows immediately afterwards.

We normalized the fitted time series by dividing them by the standard deviation of the residuals from the first-level GLM. This procedure emphasized voxels with large signal-to-noise ratios over voxels with low signal-to-noise ratios. The normalized time series were then averaged within all Q cortical parcels, resulting in a TxQ matrix (X) of cortical features. For the cerebellum, we modeled each of the p=6937 cerebellar voxels separately (SUIT atlas space with 3 mm resolution), resulting in a TxP data matrix (Y). The cortical data were further z-standardized (as in a partial correlation analysis) before entering them into regression models (see *Model Estimation*).

When estimating connectivity models, correlated fMRI noise across the cortex and cerebellum (e.g. head movement, physiological artifacts) can lead to the erroneous detection of connectivity between structures. This is especially problematic for the superior cerebellum and the directly abutting regions of the occipital and inferior temporal lobes. To negate the influence of any noise process that was uncorrelated with the tasks, we used a 'crossed' approach to train the models. Because there were two sessions (same tasks, different order) in each task set (A and B), we were able to predict the cerebellar fitted time series for the first session by the cortical time series from the second session, and vice-versa (see *Figure 1*).

$$Z_1 b_1 X \sim Z_1 b_2 Y \qquad (4)$$

We did this separately for each task set, and given that the sequence of tasks was randomized across sessions, we could conclude from this approach that this prediction was based on task-related signal changes rather than fluctuations attributable to noise processes given that noise is not shared across scanning sessions.

## Cortical parcels

We created a set of models that used different levels of granularity to subdivide the cortex. Specifically, each hemisphere was divided into a set of regular hexagonal parcels, using icosahedrons with 42, 162, 362, 642, or 1002 parcels per hemisphere [see (*Zhi et al., 2022*) for details]. Parcels that were completely contained within the medial wall were excluded, resulting in 80, 304, 670, 1190, or 1848 parcels combined for the left and right hemisphere. Activity within a parcel was averaged across voxels for each condition, defining the Q regressors for each model.

The regular icosahedron parcellations are arbitrary in the sense that they do not align to functional boundaries in the human neocortex more than expected by chance (*Zhi et al., 2022*). We therefore also employed functionally-defined parcellations, repeating the main analyses using 12 different parcellations derived from resting state fMRI data (see *Figure 2—figure supplement 3*).

## Model estimation

We used three regression methods to estimate the connectivity matrix $\hat{W}$ at the individual participant level. In the core analyses, each of these methods was combined with the five arbitrary cortical parcellations, resulting in 15 connectivity models. Each method was selected to favor a specific pattern of cortico-cerebellar connectivity. For the *winner-take-all* models, we assumed that the time series of each cerebellar voxel is explained by one and only one cortical region. To build this model, we simply chose the cortical region with the highest correlation with the cerebellar voxel in question. The connectivity weight corresponding to the rest of the cortical regions were set to 0.

The other two methods allowed for some degree of convergence in the cortical input to each cerebellar voxel. *Lasso regression* [least absolute shrinkage and selection operator (*Tibshirani, 1996*)], seeks to explain activity in each cerebellar voxel by a restricted set of cortical features. Specifically,

Lasso minimizes the squared-error, plus an additional penalty calculated as the sum of the absolute values (L1-norm) of the regression coefficients:

$$\hat{W} = \underset{w}{argmin} \, \|Y - XW\|_2^2 + \lambda \, \|W\|_1 \tag{5}$$

In contrast, *Ridge regression* penalizes large connectivity weights, attempting to find a broad set of cortical regions to explain each cerebellar voxel. Ridge regression minimizes the following loss function:

$$\hat{W} = \underset{w}{argmin} \, \|Y - XW\|_2^2 + \lambda \, \|W\|_2^2 \tag{6}$$

Both Lasso and Ridge models include a hyperparameter $\lambda$ that must be tuned in model estimation. The value of these regularization parameters was determined using gridsearch with ourfold cross-validation on the training set. We divided the conditions of the training set into four non-overlapping subsets, reconstructed the time series using tasks from three of the four subsets and evaluated the model using the left-out subset. *Figure 2—figure supplement 4* depicts the average predictive accuracy for the training set and the optimal value of $\lambda$ for the Lasso and Ridge models.

## Model testing

After model training with Task Set A, Task Set B was used as the test set. We applied the same procedure to construct the **X** and **Y** matrices. We then used the estimated weights for each model to generate predicted voxel-wise cerebellar activity patterns from the observed cortical time series. These predictions were generated at each level of granularity for each participant. Model performance was measured by correlating predicted and observed cerebellar time series for each voxel. As with the procedure for model training, the cortical and cerebellar time series were crossed: Cortical time series from session 3 were used to predict cerebellar time series from session 4 and vice versa. Model predictions were calculated separately for each cerebellar voxel and visualized on voxelwise maps of the cerebellar cortex.

In sum, we evaluated 15 models in which the cortical parcels were based on an arbitrary icosahedron, based on the combination of three regression methods and five levels of granularity. For the 12 functionally-defined parcellations (*Figure 2—figure supplement 3*), we limited the evaluation to the WTA and Ridge models.

## Noise ceiling

For each model evaluated in the current study, the noise ceiling quantifies the expected performance of each model, under the assumption that the estimated weights reflect the true connectivity between the neocortex and cerebellum. When predicting the cerebellar activity patterns, our models are limited by two factors: Measurement noise associated with the cerebellar data and measurement noise associated with the cortical data.

To estimate these sources of noise, we used the fitted cerebellar time-series for the two sessions, assuming that each session's data are composed of the true time series Y* plus noise ε: $Y_i = Y^* + \epsilon$. If $X_i W$ signifies the variance of the true time series, and $\sigma_\epsilon^2$ the variance of the noise, then the correlation between the two sessions, i.e. the split-half reliability, is:

$$r\left(\mathbf{Y}_1, \mathbf{Y}_2\right) = \frac{\sigma_Y^2}{\sigma_Y^2 + \sigma_\epsilon^2}. \tag{7}$$

Similarly, the reliability of the prediction, even if we knew the true connectivity weights W, is limited by the noise on the cortical data X. The prediction $X_i W$ has noise variance, $\sigma_{XW}^2$ which can be estimated by calculating the split-half reliability across the two sessions:

$$r(X_1 W, X_2 W) = \frac{\sigma_Y^2}{\sigma_Y^2 + \sigma_{XW}^2} \tag{8}$$

Thus, for the true model the expected correlation would be:

$$r_{ceil} = r(X_i W, Y_i) = \frac{\sigma_Y^2}{\sqrt{\sigma_Y^2 + \sigma_{XW}^2}\sqrt{\sigma_Y^2 + \sigma_\epsilon^2}} = \sqrt{r(Y_1, Y_2)r(X_1 W, X_2 W)} \tag{9}$$

Because we do not know the true weights, we use the estimated weights from each model to estimate the noise ceiling. Thus, the noise ceiling is model dependent, specifying what the correlation would be if the current model was the true model.

## Cortico-cerebellar convergence

Two measures were used to assess the amount of convergence of cortical inputs to each cerebellar voxel. For the first measure, we calculated the percentage of the cortical surface contributing to the prediction in each cerebellar voxel. We used the estimated weights from the Lasso regression model with the best unnormalized performance (80 cortical parcels). We opted to focus on the Lasso model since it sets weights that make a negligible contribution to the predictions to zero, thus identifying the sparsest cortical input that explains the cerebellar data. To determine the likely input area, we determined the number of non-zero coefficients for each cerebellar voxel, and expressed this as a percentage of the cortical surface. We also repeated these analyses using Ridge regression (*Figure 3—figure supplement 1*). Because Ridge does not result in zero coefficients, we applied a threshold, counting the number of connectivity weights with a value one standard above the mean of the weight matrix.

For the second measure, we calculated the dispersion (spherical variance) of the input weights. Using the spherical representation of the cerebral hemispheres from FreeSurfer, we defined unit vectors, $v_i$, pointing from the center of the sphere to the center of each of the cortical parcels. Each vector was then weighted by the cortico-cerebellar connectivity weight for that region $w_i$. All negative connectivity weights were set to zero. The spherical variance for hemisphere h across all $Q_h$ parcels of this hemisphere was defined as:

$$var_h = 1 - \left| \frac{1}{Q_h} \sum_{Q_h} w_i v_i \right| \tag{10}$$

To obtain a composite measure, we averaged the variances for the two hemispheres, with each weighted by the size of the summed (non-negative) connectivity weights. For summary and statistical testing, we then averaged the size of the input area, as well as the dispersion, across cerebellar voxels within each of the 10 functionally defined MDTB regions.

## Model recovery simulations

Even though we are using data from a broad task battery, there is substantial collinearity between the cortical parcels. This raises the questions of whether our approach can reliably distinguish between models of sparse and broad connectivity, and whether we can detect differences in convergence across regions in an unbiased fashion.

To validate our model-selection approach we conducted a series of model recovery simulations. Using the observed cortical activity patterns from each individual participant (with 80–1848 parcels), we simulated artificial cerebellar data for 2000 voxels. The artificial connectivity weights were chosen under two scenarios. To simulate one-to-one connectivity, each cerebellar voxel was connected to only one randomly chosen cortical parcel. To simulate broad convergence, the connectivity weights were drawn from a normal distribution with mean = 0, and a variance of 0.2. We then generated artificial data following Equation 2, both for the training data (using the cortical data from Task Set A) and for the testing data (using the cortical data from Task Set B). The measurement noise (E) was drawn from a normal distribution with mean = 0. The variance (0.25) was adjusted to approximate the signal-to-noise level observed in the empirical data. Model estimation (including hyperparameter-tuning) and model testing were then conducted following the procedure used for the empirical data (see above).

We also tested whether differences in cortical input area (*Figure 4b*) could be due to differences in collinearity in the associated cortical parcels. To address this question, we conducted a simulation with 80 cortical parcels. For each MDTB region and participant, we simulated cerebellar data using one-to-one connectivity with the cortical parcel that had the highest average lasso-regression coefficient for that region and participant. Thus, we replaced the average activity profile in each cerebellar region by the profile of the most similar cortical parcel. We submitted these data to a Lasso regression with

log-lambda set to –3 and calculated the percentage of the cortical surface with non-zero connectivity weights in the same manner as done with the real data.

### Generalization to new participants

Although the models were trained and tested on distinct datasets with more than half the tasks unique to each set, the data for training and testing were obtained from the same set of participants. As a stronger test of generalization, we evaluated the models with data from a new experiment involving naive participants (n=20, 11 females, 9 males, mean age = 21.3). These participants were trained to perform a 5-task battery (cognitive control, social prediction, action prediction, visual search, semantic prediction). For each task, there was an easy and hard condition (e.g., 2-back vs 0-back for cognitive control, high vs low predictability), along with the task's associated instruction. Thus, there were a total of 16 conditions when including rest (fixation). Each participant completed five sessions, with fMRI data obtained in the first, third, and fifth sessions. Within a scanning session, there were six 11-min runs, and each run included three repetitions of each task (35 s) with a 10 s rest period prior to each 5 s instruction period.

The participants were scanned on a Siemens MAGNETOM TrioTim syngo MR B17 located in the Henry Wheeler Jr. Brain Imaging Center at the University of California, Berkeley. The protocol used the following parameters: 1 s repetition time; field-of-view measuring 20.8 cm; A-to-P phase encoding direction; 48 slices; 3 mm thickness; in-plane resolution 2.5×2.5 mm$^2$. To localize and normalize the functional data, a high-resolution, whole-brain anatomical scan (T1-weighted MPRAGE, 1 mm isotropic resolution) was acquired during the first session. fMRIPrep (*Esteban et al., 2019*; *Esteban et al., 2020*; https://fmriprep.org/en/stable/) was used to preprocess the anatomical and functional data, following the same analysis procedures as conducted for the main experiment.

To generate estimates of activity (beta weights), a General Linear Model (GLM) was fitted for each voxel to the time series data. Each of the 16 conditions was modeled as a separate regressor, with this repeated separately for each imaging run. Nipype (https://nipype.readthedocs.io/en/latest/) and custom-written scripts in Python were used to estimate the beta weights from a first-level GLM.

To evaluate model generalization to this new dataset, we used the full set of cortico-cerebellar connectivity models estimated with the original data set, limiting the analysis to the regular cortical parcellations. Since the participants were different from the original data set, we averaged the model weights for the original 24 participants (i.e. 3 model types x 5 levels of granularity). Each model was then used to predict cerebellar activity from the cortical activity obtained in the new study, with these predictions compared to actual cerebellar activity at the individual level. Given that the experiment was a training study, we expected that both the true activation patterns, as well as the signal-to-noise ratio would change across sessions. We therefore determined the noise ceiling for each session separately by calculating the reliability across odd and even runs within each session. The normalized predictive accuracy was determined separately for each of the three imaging sessions, with the results across sessions averaged.

### Source data files

Model training ('model_training_ridge.csv' and 'model_training_lasso.csv') and evaluation ('model_evaluation.csv') summary results are included as additional source data in the manuscript.

## Acknowledgements

This work was supported by the Canadian Institutes of Health Research (PJT 159520 to JD), the Canada First Research Excellence Fund (BrainsCAN to Western University), and the National Institute of Health (NS092079, NS105839 to RBI). Thanks to Da Zhi for providing cerebral cortical parcellations.

## Additional information

#### Competing interests

Richard B Ivry: is a co-founder with equity in Magnetic Tides, Inc. Jörn Diedrichsen: Reviewing editor, eLife. The other authors declare that no competing interests exist.

## Funding

| Funder | Grant reference number | Author |
|---|---|---|
| Canadian Institutes of Health Research | PJT 159520 | Jörn Diedrichsen |
| National Institutes of Health | NS092079 | Richard B Ivry |
| National Institutes of Health | NS105839 | Richard B Ivry |

The funders had no role in study design, data collection and interpretation, or the decision to submit the work for publication.

## Author contributions

Maedbh King, Conceptualization, Data curation, Software, Formal analysis, Validation, Visualization, Methodology, Writing – original draft, Project administration, Writing – review and editing; Ladan Shahshahani, Software, Formal analysis, Validation, Visualization, Methodology, Writing – original draft, Project administration, Writing – review and editing; Richard B Ivry, Conceptualization, Resources, Supervision, Funding acquisition, Investigation, Writing – original draft, Writing – review and editing; Jörn Diedrichsen, Conceptualization, Resources, Data curation, Software, Formal analysis, Supervision, Funding acquisition, Investigation, Methodology, Writing – original draft, Project administration, Writing – review and editing

## Author ORCIDs

Maedbh King ⓘ http://orcid.org/0000-0001-5374-1011
Ladan Shahshahani ⓘ http://orcid.org/0000-0001-6189-9994
Richard B Ivry ⓘ http://orcid.org/0000-0003-4728-5130
Jörn Diedrichsen ⓘ http://orcid.org/0000-0003-0264-8532

## Ethics

Human subjects: All participants gave informed consent under an experimental protocol approved by the institutional review board at Western University (Protocol number: 107293). Undergraduate and graduate students were recruited (via posters) from the larger student body at Western University. The sample was biased towards relatively high-functioning, healthy and young individuals. The final sample consisted of 24 healthy, right-handed individuals (16 females, 8 males; mean age=23.8 years old, SD=2.6) with no self-reported history of neurological or psychiatric illness.

## Decision letter and Author response

Decision letter https://doi.org/10.7554/eLife.81511.sa1
Author response https://doi.org/10.7554/eLife.81511.sa2

# Additional files

## Supplementary files

• MDAR checklist

## Data availability

The multi-domain task battery (MDTB) published in *King et al., 2019* was analyzed in the current study. The raw behavioral and whole-brain imaging data generated by this task battery are available on the openneuro data sharing repository (https://openneuro.org/datasets/ds002105/share). The cortical parcellations used in the current study are available to download from https://github.com/DiedrichsenLab/fs_LR_32/tree/master (copy archived at *Diedrichsen et al., 2022*) and the cerebellar parcellations and contrast maps are available to download from https://www.diedrichsenlab.org/imaging/mdtb.htm. The cerebellar maps are also available for interactive visualization in an atlas viewer (https://www.diedrichsenlab.org/imaging/AtlasViewer/). The experimental code is publicly available at https://github.com/maedbhk/cerebellum_connectivity (copy archived at *King, 2023*). Model training and evaluation summary results are included as additional source data in the manuscript.

The following previously published dataset was used:

| Author(s) | Year | Dataset title | Dataset URL | Database and Identifier |
|---|---|---|---|---|
| King M, Hernandez-Castillo C, Poldrack RA, Ivry RBI, Diedrichsen J | 2019 | Multi-Domain Task Battery (MDTB) | https://openneuro.org/datasets/ds002105/share | OpenNeuro, ds002105 |

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
