## [Editor Report]

This study presents the valuable finding that the human cerebellum receives highly convergent (rather than sparse) task-related connectivity from the cortex. A compelling case is made that this convergent connectivity supports a wide variety of cognitive functions, though it will be important for future work to fully verify the cortical-to-cerebellar directionality of the results. The work will be of broad interest to cognitive neuroscientists interested in the role of connectivity in cognition.

---

## [Decision Letter]

**Decision letter after peer review:**

Thank you for submitting your article "A task-general connectivity model reveals variation in convergence of cortical inputs to functional regions of the cerebellum" for consideration by *eLife*. Your article has been reviewed by 3 peer reviewers, one of whom is a member of our Board of Reviewing Editors, and the evaluation has been overseen by Chris Baker as the Senior Editor. The reviewers have opted to remain anonymous.

Essential revisions:

1) Claims of directionality are overstated and should be revised accordingly.

2) The authors should address the differences in model complexity, ideally through an additional analysis (e.g., involving AIC or BIC).

3) More details on the connectivity method used, especially how it handles the task-evoked variance inflation problem.

4) The authors should discuss the possibility that motor tasks are inherently simpler than non-motor tasks, which may have biased their results regarding motor vs. non-motor regions.

*Reviewer #2 (Recommendations for the authors):*

The authors should compare model performance while accounting for model complexity. One solution is for the Lasso/Ridge models to have the "WTA" weights to be fixed while randomly permuting all other weights. If the Ridge model is truly better, you would expect this random model to perform similarly to the WTA model but worse than the original Ridge/Lasso models. This would suggest that more inputs do not significantly explain more variances in the cerebellum just by chance.

The authors should expand the discussion on the closed-loop anatomical relationship between the cerebellum and the cortex and how its multi-synaptic relationship can allow converging inputs from the cortex to the cerebellum.

For figures 3 and 4, it would be nice to expand and show physically where and how widespread those cortical inputs are. That can help readers contextualize the physical level of "convergence" of the cortex onto cerebellar regions.

Instead of interpreting the results as cortico-cerebellar inputs, discuss how they can be bidirectional interactions.

*Reviewer #3 (Recommendations for the authors):*

– The authors note that a "convergent architecture would suggest that subregions in the cerebellum integrate information across disparate cortical regions and may coordinate their interactions." It would be interesting to evaluate the degree to which the "disparate" regions that may provide convergent inputs to the cerebellum, as identified in this study, are functionally/anatomically related.

– The authors discuss findings from Pisano et al. 2021 as anatomical evidence for cerebellar outputs from specific regions reaching a variety of cortical targets. However, it is important to note that in most species, and especially in the mouse, it is quite unlikely that any one anatomical tracer injection into the cerebellar cortex can target a functionally distinct region of the cerebellum. See, for example, Fujita et al. 2010 J Comp Neurol where small injections into marmoset cerebellar cortex are difficult to localize to specific longitudinal bands in the cerebellar cortex.

– A map of the cerebellum showing the MDTB regions would be useful to include as a guideline in Figure 3

[Editors' note: further revisions were suggested prior to acceptance, as described below.]

Thank you for resubmitting your work entitled "A task-general connectivity model reveals variation in convergence of cortical inputs to functional regions of the cerebellum" for further consideration by *eLife*. Your revised article has been evaluated by Chris Baker (Senior Editor) and a Reviewing Editor.

The manuscript has been improved but there are some remaining issues that need to be addressed, as outlined below:

*Reviewer #1 (Recommendations for the authors):*

The authors did an excellent job improving their manuscript based on previous reviews, except for one major point described below.

R1 point 3 was responded to, but not meaningfully addressed. The authors seemed to misunderstand the issue that regressing out mean task-evoked activity is meant to deal with, resulting in the authors effectively ignoring the issue with regard to their own study. Indicative of misunderstanding the issue, the authors state: "Returning to the papers cited by the reviewer, these are designed to look at connectivity not related to the task-evoked activity." This is a common misunderstanding in the field: Rather than looking at connectivity not related to task-evoked activity, regressing out mean task-evoked activity is meant to deal with a causal confound (common inputs from stimuli), while still allowing estimation of connectivity related to the task-evoked activity. This interest in the impact of the task-related activity on connectivity is clear from the simple logic of most studies utilizing mean task-evoked activity regression, wherein connectivity estimated in the context of condition-specific task-evoked activity is compared. (If all task-evoked activity were removed then there would be no point in comparing task conditions.) What is left over for estimating functional connectivity estimation is trial-by-trial (and moment-to-moment) variation in task-evoked activity (along with spontaneous activity). So, the authors have not dealt with the possibility that common stimulus inputs – such as centrally presented visual stimuli simultaneously activating left and right hemispheres, generating false inter-hemispheric functional connections – are driving some of their connectivity estimates. Ideally, this confound would be dealt with, but at the very least the authors should acknowledge this limitation of their methods.

Beyond this issue (and perhaps more importantly), it is difficult to reconcile the present study's approach with more standard task-related functional connectivity approaches. How can connectivity values estimated via the present study's approach be interpreted? How do these interpretations differ from those of more common task-related functional connectivity approaches? Is there enough advantage to using the present study's unconventional task-related functional connectivity method to make it worth the extra effort for readers to make sense of it and learn to relate it to more standard approaches? The presence of all of these unanswered questions makes it difficult to properly evaluate the manuscript and its results.

Further, in my attempt to answer the above questions myself, I have read and re-read the description of the connectivity approach, and yet I have not been able to make sense of it. I doubt this is my own lack of knowledge or intelligence, but even if that were the case it is likely that many (perhaps most) readers would have the same issue. Therefore, I recommend rewriting the "Connectivity Models" section (within "Methods") to include more details. Some specific details that would be helpful to include:

– The meaning of "fitted time series" is ambiguous. Does this mean the original time series multiplied by the fit β estimate for each condition, resulting in the original time series modulated by the fit to the task regressor for each condition? Or does this mean the regressor time series multiplied by the fit β estimate for each condition? If it is the first option I can imagine interpreting the resulting functional connectivity estimates in a fairly similar way to standard task-related functional connectivity. However, if it is the second option, then I think the interpretation would be quite different from standard task-related functional connectivity. In such a case I think much more details on how to interpret the resulting connectivity values would be warranted. Of course, it could be a third option I haven't considered, in which case it would be helpful to have that option detailed.

– It is unclear what this detail means (pg. 24): "This procedure reweighted the activity estimates in an optimal manner, accounting for the fact that estimates for the instructions were based on 5 s of fMRI data, while estimates for the conditions were based on 10-30 s of data." In what sense is this accounting for the duration of task conditions? This makes me wonder whether the connectivity estimates were based on using cortical β estimates (rather than time series) to predict cerebellar β estimates (rather than time series), which could account for condition duration quite well. But the description is ambiguous with respect to these kind of details, and this possibility doesn't seem to fit well with earlier descriptions of the method.

– It is unclear what this detail means (pg. 24): "This method also mitigates problems that may arise in event-related designs due to correlations between regressors (Mumford et al., 2012)." How exactly? Also, what exactly are the problems that may arise in event-related designs due to correlations between regressors? It sounds like this is related to collinearity, but it is unclear which aspects of the method help with collinearity.

– The details regarding the "crossed" approach need to be fleshed out. It is stated (pg. 25): "we used a "crossed" approach to train the models: The cerebellar fitted time series for the first session was predicted by the cortical time series from the second session, and vice-versa". This procedure is also presented in an ambiguous manner in Figure 1. Right now it is stated in the manuscript that the session 2 cortical time series are used to predict the session 1 cerebellar time series, and yet it is unclear how that would be possible when there are different task conditions across the sessions. Is the connectivity model built using both cortical and cerebellar time series during session 2 (betas from cortex predicting cerebellum), then both cortical and cerebellar time series are used from session 1 to test the session 2 connectivity model (session 2 betas multiplied by session 1 cortical time series to predict session 1 cerebellar time series)? Put more simply: is it the case that the session 2 connectivity model is tested using session 1 data, and that the session 1 connectivity model is tested using session 2 data? If so, the description should change to better describe this. In any case, the description should be clarified with more details.

---

## [Author Response]

Essential revisions:1) Claims of directionality are overstated and should be revised accordingly.

We agree that the correlation structure of BOLD signals in the neocortex and cerebellum is shaped by the closed-loop (bi-directional) interactions between the two structures. As such, some of the observed convergence could be caused by divergence of cerebellar output. We have added a new section to the discussion on the directionality of the model (Page 18).

That said, there are strong reasons to believe that our results are mainly determined by how the neocortex sends signals to the cerebellum, and not vice versa. An increasing body of physiological studies (and this includes newer papers, see response to reviewer #1, comment #1 for details) show that cerebellar blood flow is determined by signal transmission from mossy fibers to granule cells and parallel fibers, followed by Nitric oxide signaling from molecular layer interneurons. Importantly, it is clear that Purkinje cells, the only output cell of the cerebellar cortex, are not reflected in the BOLD signal from the cerebellar cortex. (We also note that *increases* in the firing rate of inhibitory Purkinje cells means *less* activation of the neocortex). Thus, while we acknowledge that cerebellar-cortical connectivity likely plays a role in the correlations we observed, we cannot use fMRI observations from the cerebellar cortex and neocortex to draw conclusions about cerebellar-cortical connectivity. To do so we would need to measure activity in the deep cerebellar nuclei (and likely thalamus).

The situation is different when considering the other direction (cortico-cerebellar connections). Here we have the advantage that the cerebellar BOLD signal is mostly determined by the mossy fiber input which, at least for the human cerebellum, comes overwhelmingly from cortical sources. On the neocortical side, the story is admittedly less clear: The cortical BOLD signal is likely determined by a mixture of incoming signals from the thalamus (which mixes inputs from the basal ganglia and cerebellum), subcortex, other cortical areas, and local cortical inputs (e.g., across layers). While the cortical BOLD signal (in contrast to the cerebellum) also reflects the firing rate of output cells, not all output cells will send collaterals to the pontine nuclei. These caveats are now clearly expressed in the Discussion section2.

On balance, there is an asymmetry: Cerebellar BOLD signal is dominated by neocortical input without contribution from the output (Purkinje) cells. Neocortical BOLD signal reflects a mixture of many inputs (with the cerebellar input making a small contribution) and cortical output firing. This asymmetry means that the observed correlation structure between cortical and cerebellar BOLD activity (the determinant of the estimated connectivity weights) will be determined more directly by cortico-cerebellar connections than by cerebellar-cortical connections. Given this, we have left the title and abstract largely the same, but have tempered the strength of the claim by discussing the influence of connectivity in the opposite direction.

2) The authors should address the differences in model complexity, ideally through an additional analysis (e.g., involving AIC or BIC).

We apologize for not having been clearer on the topic of model complexity in the original submission. Our approach (Figure 1) is designed to compare models of different complexity in an unbiased fashion. Parameters for all models were estimated on the training set (Task set A, Figure 1A). This includes the large number of connectivity weights (W_hat) and the single regularization coefficient for Ridge and Lasso regression. The models were then tested on the data from two independent imaging sessions (Task set B, Figure 1B). Using independent training and test data sets is the gold-standard in statistical model comparison (Stone, 1974), and fully corrects for model complexity i.e., any model that overfits the data will, on average, perform worse on an independent test dataset (i.e., Hastie et al. 2009, Chapter 7).

Training and testing was performed within-subjects to account for the considerable inter-individual variation in cortical and cerebellar functional organization. However, to test how well each model generalizes to new subjects, we also averaged the estimated weights across subjects and re-tested the model on a dataset with new subjects performing a new set of tasks (Figure 1C).

AIC is a well-established approximation to the expected cross-validation error in the setting of Gaussian linear regression (Stone 1977). While extensions to WTA (i.e. best-subset selection), Ridge regression and Lasso regression exist, the comparison of models across these approaches using AIC has, to our knowledge, not been well-validated. Thus, we think that using separate training and test sets is the safest and most transparent way to compare models with different structures.

Another important reason to compare models based on their ability to predict independent datasets, rather than using AIC, is that we can assess how well the models generalize to new task conditions. AIC (and related methods) approximates the expected cross-validation error in predicting Y (the cerebellar activity patterns) given a fixed X (the cortical activity patterns). However, in our case the number of tasks is smaller than the number of cortical parcels. Therefore, there will be strong linear dependencies between the different cortical parcels, such that a model may associate a cerebellar region with the wrong combination of cortical parcels (see pt 3). To protect against this, we really want to test on a new random X (e.g., on new tasks). New task sets will lead to somewhat different co-dependencies between cortical regions, and a model that learned the wrong combination of cortical parcels should perform poorly on these new (random) X’s, but not on the old (fixed) X’s. We therefore think that the out-of-sample test on a new taskset is a much stronger criterion for model comparison.

We now point this logic out more clearly in the Introduction and Results. We hope this is a more appropriate and convincing way of dealing with the problem of model comparison than AIC.

References:

Hastie, T., Tibshirani, R., and Friedman, J. (2009). *The Elements of Statistical Learning*. Second Edition.

Stone, M. (1974). Cross-validatory choice and assessment of statistical predictions, Journal of the Royal Statistical Society Series B 36: 111–147.

Stone, M. (1977). An asymptotic equivalence of choice of model by cross validation and Akaike’s criterion, Journal of the Royal Statistical Society Series B. 39: 44–7.

3) More details on the connectivity method used, especially how it handles the task-evoked variance inflation problem.

We have expanded the description of our connectivity approach in the revised manuscript. It is important to stress that our approach only relied on task-evoked activity. Rather than removing the task-related activity from the time series (as for example, Cole et al., 2019) or using the entire time series, we used only the fitted (and hence task-evoked) activity. All other fluctuations were removed using the crossed approach of predicting cerebellar activity from one half of the data based on cortical activity from the other half of the data for each task set (see response to reviewer #1, comment #4). Models estimated based on the fitted (task-evoked) activity were able to predict the cerebellar activity patterns in Task Set B better than models based on the entire time series or models based on the residuals after removing task-evoked activity.

A task-based approach comes with two important concerns: First, for a small number of tasks, collinearity between different cortical areas may prevent us from determining the correct connectivity model. Second, the particular choice of tasks may bias the estimate of connectivity, and the models may not generalize well to other tasks.

In this paper, we addressed these problems by deliberately choosing a very broad task-set, one designed to activate different cortical areas as independently as possible. Additionally, we evaluated connectivity models in how well they predicted activity patterns from tasks that were not included in task set A (see also pt 2) – this ensures that we pick a model that generalizes well.

To determine how successful we were in picking tasks that allowed us to estimate a comprehensive connectivity model, we calculated the variance-inflation-factor (VIF) for each cortical parcel for the Ridge regression model (Marquardt, 1970). The VIF tells us how variable our estimate of the connectivity weight to a particular cortical parcel is due to collinearity with other cortical parcels. All cortical activity profile (columns of **X**) were z-standardized before entering into Ridge regression, such that:VIF=(XTX+Iα)−1XTX(XTX+Iα)−1

Author response image 1 shows the average VIF for all 1848 cortical parcels (log-α = 8). The collinearity was most severe in anterior and inferior temporal regions, orbitofrontal cortex and some regions of the insula. This indicates that our task battery did not systematically activate and/or dissociate activity in these regions. Importantly, however, this limitation is not observed in the rest of the brain. The VIF in visual, motor, and associative regions was fairly uniform. Thus, the differences in convergence that we found between hand-motor regions and more cognitive regions of the cerebellum does not appear to be caused by different degrees of collinearity in these regions.

**Author response image 1. sa2fig1:** 

To further address this issue, we conducted an additional simulation analysis using the Lasso model and 80 cortical parcels (as in Figure 4b). For each MDTB region and participant, we simulated artificial cerebellar data using the activity profile from each cortical parcel that was most similar to that parcel and added Gaussian noise (see methods, model recovery simulation). We then estimated a connectivity model (Lasso, log λ = -3, 80 parcels) and calculated the percentage of cortical area with non-zero connectivity weights.As can be seen from Author response image 2, the cerebellar regions showed comparable input areas. Thus, the increased convergence observed for cerebellar region 3-10 was not an artifact of the collinearity of the cortical regressors.

References:Cole, Michael W., Takuya Ito, Douglas Schultz, Ravi Mill, Richard Chen, and Carrisa Cocuzza. 2019. "Task Activations Produce Spurious but Systematic Inflation of Task Functional Connectivity Estimates." NeuroImage 189 (April): 1-18.

Marquardt, D. W. (1970). Generalized Inverses, Ridge Regression, Biased Linear Estimation, and Nonlinear Estimation. *Technometrics*, *12*(3), 591. https://doi.org/10.2307/1267205

4) The authors should discuss the possibility that motor tasks are inherently simpler than non-motor tasks, which may have biased their results regarding motor vs. non-motor regions.

We do not believe that the difference in convergence between hand-motor and cognitive regions of the cerebellum are biased by complexity differences between motor and non-motor tasks. As we discuss in pt. 3, almost all regions of the cortex (including primary, premotor, and supplementary motor regions) were systematically activated and dissociated by our broadband set of tasks.

The above is an empirical argument. Theoretically we recognize there are many ways of defining complexity, a non-trivial exercise. We included two basic motor tasks in our battery, simple finger tapping and finger sequence tasks. Both activated primary and somatosensory motor cortices (as did most of our cognitive tasks since many required a left- or right-hand response). The main difference between the two motor tasks was that the sequence task elicited higher activation across these regions, as well as elicited activation in premotor and parietal areas. We do think it will be interesting in future research to include a broader set of motor tasks (e.g., include foot and tongue movements). Nonetheless, we do think we have a reasonable manipulation of complexity in the motor domain and yet, still find a difference in convergence between the motor and cognitive regions. Moreover, based on our statistical analysis of collinearity across the cortex (pt. 3), we show that differences in convergence between motor and non-motor regions cannot be attributed to different degrees of collinearity in these regions.

Reviewer #2 (Recommendations for the authors):The authors should compare model performance while accounting for model complexity. One solution is for the Lasso/Ridge models to have the "WTA" weights to be fixed while randomly permuting all other weights. If the Ridge model is truly better, you would expect this random model to perform similarly to the WTA model but worse than the original Ridge/Lasso models. This would suggest that more inputs do not significantly explain more variances in the cerebellum just by chance.

Our approach strongly controls for the differences in complexity between the WTA, Lasso and Ridge models. We elaborate on this in our response to General Comment #2.

The authors should expand the discussion on the closed-loop anatomical relationship between the cerebellum and the cortex and how its multi-synaptic relationship can allow converging inputs from the cortex to the cerebellum.

This is an important point. We have expanded on this in the revised Discussion section and provide an extended response in General Comment #1.

For figures 3 and 4, it would be nice to expand and show physically where and how widespread those cortical inputs are. That can help readers contextualize the physical level of "convergence" of the cortex onto cerebellar regions.

As noted above, we have reversed the order of Figures 3 and 4, using the examples to now set the stage for the quantitative metrics concerning “physical spread” of cortical inputs. We hope that this reordering provides a clearer picture of connectivity between cortex and cerebellar regions, and allows for a better contextualization of the dispersion metrics used to quantify convergence.

Instead of interpreting the results as cortico-cerebellar inputs, discuss how they can be bidirectional interactions.

We have added an extended discussion of this important point (page 18, see response to General Comment #1).

Reviewer #3 (Recommendations for the authors):– The authors note that a "convergent architecture would suggest that subregions in the cerebellum integrate information across disparate cortical regions and may coordinate their interactions." It would be interesting to evaluate the degree to which the "disparate" regions that may provide convergent inputs to the cerebellum, as identified in this study, are functionally/anatomically related.

This is an interesting issue. Many cerebellar regions appear to receive convergent inputs from disparate cortical regions that are functionally and/or anatomically connected. For example:

MDTB Region 3 (oculomotor vermis) was best explained by a large and relatively dispersed set of cortical regions, including intraparietal sulcus, the frontal eye fields (FEF), and extrastriate visual areas. There are established anatomical connections between FEF and extrastriate visual regions in the macaque (Schall et al. 1995). Moreover, neurostimulation studies have demonstrated that activity in the human FEF has a direct effect on sensitivity of extrastriate visual regions, suggesting a causal relationship between these regions (Silvanto et al. 2006).MDTB regions 7 and 8, which include left and right lobule IX and Crus I/11 respectively, are explained by a relatively restricted set of cortical regions, many of which overlap with subregions of the default mode network (angular gyrus; middle temporal cortex, middle frontal gyrus and inferior frontal gyrus; Smallwood et al. 2021). Previous work has suggested a similar relationship between lobule IX and subregions of the DMN (Guell et al. 2018). Further, based on our previous work, we find that this combination of cerebellar regions is often activated in DMN-like tasks such as rest and movie watching (King et al. 2019).

References:

Schall, J. D., Morel, A., King, D. J., and Bullier, J. (1995). Topography of visual cortex connections with frontal eye field in macaque: convergence and segregation of processing streams. *Journal of Neuroscience*, *15*(6), 4464-4487.

Silvanto, J., Lavie, N., and Walsh, V. (2006). Stimulation of the human frontal eye fields modulates sensitivity of extrastriate visual cortex. *Journal of neurophysiology*, *96*(2), 941-945.

Smallwood, J., Bernhardt, B. C., Leech, R., Bzdok, D., Jefferies, E., and Margulies, D. S. (2021). The default mode network in cognition: a topographical perspective. *Nature reviews neuroscience*, *22*(8), 503-513.

Guell, X., Schmahmann, J. D., Gabrieli, J. D., and Ghosh, S. S. (2018). Functional gradients of the cerebellum. *ELife*, *7*, e36652.

King, M., Hernandez-Castillo, C. R., Poldrack, R. A., Ivry, R. B., and Diedrichsen, J. (2019). Functional boundaries in the human cerebellum revealed by a multi-domain task battery. *Nature neuroscience*, *22*(8), 1371-1378.

– The authors discuss findings from Pisano et al. 2021 as anatomical evidence for cerebellar outputs from specific regions reaching a variety of cortical targets. However, it is important to note that in most species, and especially in the mouse, it is quite unlikely that any one anatomical tracer injection into the cerebellar cortex can target a functionally distinct region of the cerebellum. See, for example, Fujita et al. 2010 J Comp Neurol where small injections into marmoset cerebellar cortex are difficult to localize to specific longitudinal bands in the cerebellar cortex.

We appreciate the reviewer’s point that tracing methods have their own limits in terms of drawing conclusions about connectivity, and thank the reviewer for pointing us to the Fujita paper as providing an example of the challenges in targeting an injection site. In one sense, these limitations make an argument for the utility of our approach -- namely, to use functional data as a way to look at the convergence question (especially since this method makes no claims about whether the cortical input is direct) (via the pons) or indirect (via projections to other cortical areas and then to the pons). We have opted to not make any changes to the manuscript with respect to this point since it seemed secondary to the inferences we make in this paper.

– A map of the cerebellum showing the MDTB regions would be useful to include as a guideline in Figure 3

As discussed in our response to Reviewer 2, we have reversed the order of Figures 3 and 4 in the revised manuscript. We did this to provide examples of differences in the physical spread of cortical regions before turning to the quantitative analysis shown in Figure 4 (formerly Figure 3). Given that the MDTB regions are included in Figure 3, the reader is now provided with this map as a guideline for Figure 4.

[Editors' note: further revisions were suggested prior to acceptance, as described below.]

The manuscript has been improved but there are some remaining issues that need to be addressed, as outlined below:Reviewer #1 (Recommendations for the authors):The authors did an excellent job improving their manuscript based on previous reviews, except for one major point described below.R1 point 3 was responded to, but not meaningfully addressed. The authors seemed to misunderstand the issue that regressing out mean task-evoked activity is meant to deal with, resulting in the authors effectively ignoring the issue with regard to their own study. Indicative of misunderstanding the issue, the authors state: "Returning to the papers cited by the reviewer, these are designed to look at connectivity not related to the task-evoked activity." This is a common misunderstanding in the field: Rather than looking at connectivity not related to task-evoked activity, regressing out mean task-evoked activity is meant to deal with a causal confound (common inputs from stimuli), while still allowing estimation of connectivity related to the task-evoked activity. This interest in the impact of the task-related activity on connectivity is clear from the simple logic of most studies utilizing mean task-evoked activity regression, wherein connectivity estimated in the context of condition-specific task-evoked activity is compared. (If all task-evoked activity were removed then there would be no point in comparing task conditions.) What is left over for estimating functional connectivity estimation is trial-by-trial (and moment-to-moment) variation in task-evoked activity (along with spontaneous activity). So, the authors have not dealt with the possibility that common stimulus inputs – such as centrally presented visual stimuli simultaneously activating left and right hemispheres, generating false inter-hemispheric functional connections – are driving some of their connectivity estimates. Ideally, this confound would be dealt with, but at the very least the authors should acknowledge this limitation of their methods.

We apologize for the confusion caused by our response. When we said “task-evoked” activity in our last response, we meant the averaged (across repeated trials) task-evoked activity, not the trial-by-trial fluctuations in the task evoked activity that you are talking about here. The connectivity model training and evaluation in our paper is relying exclusively on this averaged task-evoked activity (see below).

As you state, this of course raises the immediate concern that a high connectivity weight between region A and B could be caused by a task that activates both regions, without the regions being connected. For example, the verbal N-back task is always associated with responses of the left hand, so if we were estimating connectivity based on this task alone, we could find high “connectivity” between the right hand-region and the working memory region. However, our task set was designed to be very broad, optimally dissociating the activity profiles across cortical regions (see Figure R1 in our last response). For example, the task set also included an object N-back task that was associated with right-hand responses. Thus, we addressed this concern by deliberately choosing a very broad task-set. Using data from this task set should strongly reduce the influence of spurious associations. Additionally, the evaluation on an independent set of tasks ensures that connectivity models that *do* reflect spurious between-region correlations that are induced by the specific choice of tasks, are penalized and will perform worse.

Thus, our approach follows a similar logic as the papers discussed in the last response.

The standard approach is to rely on the fact that the neural fluctuations captured in the residuals (or in the trial-by-trial variations of the regression weights) are rich and varied enough, such that two areas that are not connected would not always correlate. Here, we are relying on the fact that our task set is rich and varied enough to ensure that two areas that are not connected would not always be co-activated. To be clear, both approaches do not guarantee that the estimated connectivity between region A and region B is not caused by a consistent indirect influence from a third region.

Beyond this issue (and perhaps more importantly), it is difficult to reconcile the present study's approach with more standard task-related functional connectivity approaches. How can connectivity values estimated via the present study's approach be interpreted? How do these interpretations differ from those of more common task-related functional connectivity approaches? Is there enough advantage to using the present study's unconventional task-related functional connectivity method to make it worth the extra effort for readers to make sense of it and learn to relate it to more standard approaches? The presence of all of these unanswered questions makes it difficult to properly evaluate the manuscript and its results.

Our approach indeed deviates in another aspect from other connectivity approaches. Rather than using correlation between time-series to derive a network structure, it runs a multiple regression analysis to explain the task-evoked activity in the cerebellum from the task-evoked activity from the neocortex. This approach is justified by the particular neuroanatomical and physiological properties of the system under study: (1) BOLD signal in the cerebellar cortex reflects almost exclusively the incoming input to the cerebellum, which in the human is dominated by the input from neocortex, and (2) there is very little recurrent activity and there are few association fibers between different regions of the cerebellum.

We therefore do not believe that our approach could not be used to model cortico-cortico connectivity. However, our chosen approach is perfectly suited to model the cortico-cerebellar system. We have now pointed out this difference, and the justification for our approach, more clearly in the manuscript.

Further, in my attempt to answer the above questions myself, I have read and re-read the description of the connectivity approach, and yet I have not been able to make sense of it. I doubt this is my own lack of knowledge or intelligence, but even if that were the case it is likely that many (perhaps most) readers would have the same issue. Therefore, I recommend rewriting the "Connectivity Models" section (within "Methods") to include more details. Some specific details that would be helpful to include:– The meaning of "fitted time series" is ambiguous. Does this mean the original time series multiplied by the fit β estimate for each condition, resulting in the original time series modulated by the fit to the task regressor for each condition? Or does this mean the regressor time series multiplied by the fit β estimate for each condition? If it is the first option I can imagine interpreting the resulting functional connectivity estimates in a fairly similar way to standard task-related functional connectivity. However, if it is the second option, then I think the interpretation would be quite different from standard task-related functional connectivity. In such a case I think much more details on how to interpret the resulting connectivity values would be warranted. Of course, it could be a third option I haven't considered, in which case it would be helpful to have that option detailed.– It is unclear what this detail means (pg. 24): "This procedure reweighted the activity estimates in an optimal manner, accounting for the fact that estimates for the instructions were based on 5 s of fMRI data, while estimates for the conditions were based on 10-30 s of data." In what sense is this accounting for the duration of task conditions? This makes me wonder whether the connectivity estimates were based on using cortical β estimates (rather than time series) to predict cerebellar β estimates (rather than time series), which could account for condition duration quite well. But the description is ambiguous with respect to these kind of details, and this possibility doesn't seem to fit well with earlier descriptions of the method.

Your second interpretation is correct. To be perfectly clear, the GLM decomposes the time series of each voxel (*t*) for each run (*r*) and session (*s*) into a fitted time series (Z b_sr_) and a residual time-series (r_sr_). The fitted activity can be further decomposed into the part of the time series that is explained by the activity for each condition (averaged across runs), and a part that is explained by the run-to-run variations around the mean activity for each condition:ts,r=Zs,rb¯s+Z(bs,r−b¯s)+rs,r

Where Z_s,r_ is the (n_timepoints x n_regressors) design matrix for session s and run r. b_s,r_ is the vector of activity estimates for a specific run, and <inline-graphic mimetype="image" mime-subtype="png" xlink:href="media/image1.png" /> is the average vector of β weight sacross runs for session *s*. For the connectivity analysis, we used only the first term – the part of the response that was explained by the average task response. We have now clearly pointed out in methods section (see *connectivity models*) how this approach differs diametrically from other connectivity approaches that remove the fitted time series (Z b) and use the residual time series (r) only to estimate connectivity.

– It is unclear what this detail means (pg. 24): "This method also mitigates problems that may arise in event-related designs due to correlations between regressors (Mumford et al., 2012)." How exactly? Also, what exactly are the problems that may arise in event-related designs due to correlations between regressors? It sounds like this is related to collinearity, but it is unclear which aspects of the method help with collinearity.

We apologize for not being clearer here. As you point out, our approach of using the fitted time-series (based on averaged β weights) is conceptually very similar to using the vector of averaged activity estimates to estimate the connectivity model (indeed this is what we did first). The difference is that the former approach reweights the activity estimates based on the impact the activity estimate has on the predicted time series.

This can be clearly seen when we calculate the variances and covariances between cortical and cerebellar data, which are the only quantities that matter for the connectivity model. For example, when we compute the inner product between the data for a cortical (x) and a cerebellar voxel, we have:

yTx=b¯xTZTZb¯y

If we had used only the average activity estimates (b), we would have:yTx=b¯xTb¯y

Comparing these two equations shows that the procedure simply reweights the activity estimates by Z^T^Z. The diagonal of this matrix contains the squared length of the support of the regressor (i.e. it accounts for the fact that estimates for the instructions were based on 5 s of fMRI data, while estimates for the conditions were based on 10-30 s of data).

The off-diagonal of Z^T^Z contains the covariance between the regressors. For example, due to the overlap of the instruction period and the following condition, these two regressors are positively correlated. Consequently, the activity estimates for the instruction and following condition are negatively correlated, a problem that arises especially in event-related designs (Mumford et al., 2012). This estimation uncertainty, however, is fully cancelled out when multiplying these estimates with the design matrix again.

Overall, this is a very minor and technical point, and we fear that it would confuse the reader rather than clarify our approach. We have therefore chosen to remove this point from the methods.

– The details regarding the "crossed" approach need to be fleshed out. It is stated (pg. 25): "we used a "crossed" approach to train the models: The cerebellar fitted time series for the first session was predicted by the cortical time series from the second session, and vice-versa". This procedure is also presented in an ambiguous manner in Figure 1. Right now it is stated in the manuscript that the session 2 cortical time series are used to predict the session 1 cerebellar time series, and yet it is unclear how that would be possible when there are different task conditions across the sessions. Is the connectivity model built using both cortical and cerebellar time series during session 2 (betas from cortex predicting cerebellum), then both cortical and cerebellar time series are used from session 1 to test the session 2 connectivity model (session 2 betas multiplied by session 1 cortical time series to predict session 1 cerebellar time series)? Put more simply: is it the case that the session 2 connectivity model is tested using session 1 data, and that the session 1 connectivity model is tested using session 2 data? If so, the description should change to better describe this. In any case, the description should be clarified with more details.

We have now added more detail explaining the “crossed approach” (see *Connectivity Models*). We have also made it clear throughout the document that there are two sessions (same tasks, separate days) for each task set: the same tasks are performed across sessions, with the order of the tasks randomized across sessions (this is true for both task sets). However, using the averaged activity estimates for each session, and the task-related regressors from the other session, we can predict the time series we should have observed if the task activated the brain in the same way across sessions. Therefore, it is possible to predict the cerebellar time series from one session using the cortical time series based on the activity estimates from another session.